# Excessive sleepiness in patients with psychosis: An initial investigation

**Sarah Reeve**[1]*, **Bryony Sheaves**[2,3], **Daniel Freeman**[2,3]

**1** Department of Clinical, Educational, and Health Psychology, University College London, London, United Kingdom, **2** Department of Psychiatry, Warneford Hospital, University of Oxford, Oxford, United Kingdom, **3** Oxford Health NHS Foundation Trust, Warneford Hospital, Oxford, United Kingdom

* sarah.reeve.18@ucl.ac.uk

## Abstract

Clinical experience indicates that excessive sleepiness and hypersomnia may be a common issue for patients with psychosis. Excessive sleepiness is typically ascribed to the sedating effects of antipsychotic medications but there may be other potential contributors such as sleep disorders and depression. Furthermore, the impact of excessive sleepiness itself on patients' symptoms and general wellbeing is yet to be examined. The current study reports an exploratory cross-sectional between-groups comparison of patients with early psychosis fulfilling criteria assessed in a diagnostic interview for problematic excessive sleepiness (n = 14), compared with those not reporting excessive sleepiness (n = 46). There were no differences between the groups in diagnosis, medication type, or antipsychotic medication dosage. There were no significant group differences in sleep duration. Significantly lower activity levels were found in the excessive sleepiness group. Insomnia and nightmares were common in those reporting excessive sleepiness. No significant differences were found in psychiatric symptoms, although data did indicate more severe cognitive disorganisation and grandiosity, but less severe paranoia and hallucinations, in the excessive sleepiness group. Wide confidence intervals and small sample size mean that care should be taken interpreting these results. Overall, this study indicates that excessive sleepiness may not be solely related to medication but also to low levels of activity and other sleep disorders. This is a novel finding that, if replicated, could indicate routes of intervention for this clinical issue. Future research should aim to disentangle directions of effect amongst sleepiness, mood, activity, and psychotic symptoms and investigate possible interventions for excessive sleepiness in psychosis.

## Introduction

> *"I don't want nine hours' sleep. Your life's boring enough without sleeping too much."*
>
> - patient quote [1]

Excessive sleepiness may well be relatively common in individuals with psychosis. For example, in our recent paper assessing sleep disorders in early psychosis, 23% of patients screened

**Data Availability Statement:** The ethical approval for the study (reflected in the consent form described above) included a requirement that the anonymised data be stored securely by the study team but that these data may be utilised in future ethically approved studies. Therefore the data are

available on request from contacting oxfordhealth.
OCAP@nhs.net or the corresponding author, but
are not able to be uploaded for free access given
these restrictions.

**Funding:** This research was supported by a
Medical Research Council Doctoral Studentship
and a Balliol College Dervorguilla Scholarship
(University of Oxford) to S.R., B.S. is supported by
an NIHR clinical doctoral fellowship (ICA-CDRF-
2017-03-088), and D.F. is supported by an NIHR
Research Professorship (NIHR-RP-2014-05-003).
This study was supported by the NIHR Oxford
Health Biomedical Research Centre. The views
expressed are those of the authors and not
necessarily those of the NHS, the NIHR or the
Department of Health.

**Competing interests:** The authors have declared
that no competing interests exist.

positively for an excessive sleepiness disorder [2]. These symptoms are often ascribed by patients and clinicians to sedation from antipsychotic medication [1, 3], due to the known effect of these medications on the histaminergic system [4]. However other potential causal contributors (e.g. depression, low activity) have not been adequately investigated, despite being possible targets for ameliorating sleepiness in patients. Furthermore, framing sedation as an unavoidable side effect of antipsychotic medication has inhibited research investigating the impact of excessive sleepiness itself on the psychiatric symptoms and general wellbeing of this patient group.

Sleepiness can be defined as a propensity to fall asleep (and while in this state, conscious effort may need to be made to stay awake), in contrast to fatigue which can be more closely related to physical exhaustion [5]. The 'gold standard' assessment for excessive sleepiness is the Multiple Sleep Onset Latency Test (MSLT; [6]). The MSLT assesses how long it takes an individual to fall asleep (sleep onset latency) in a quiet environment at multiple points throughout the day, with short average sleep onset (less than five minutes) being indicative of excessive sleepiness. However, the MSLT is not possible to conduct outside of specialist sleep centres, therefore screening questionnaires such as the Epworth Sleepiness Scale (ESS; [7]) are used, which provide cut-off values for problematic sleepiness. 'Hypersomnia' as a primary sleep disorder is defined as excessive sleepiness presenting alongside extended sleep duration i.e. either ≥11 hours sleep on average per 24 hour period [8], or ≥9 hours of nocturnal sleep [9], in the absence of any other explanatory mental or physical health factors. However, the term 'hypersomnia' is also used broadly to refer to self-reported difficulties with excessive sleepiness and/ or oversleeping as a secondary symptom of other sleep disorders (e.g. sleep apnea), or psychiatric disorders such as depression [9].

The relationship between this broader concept of hypersomnia and mental health has predominantly been researched in relation to depression, although even within this area it has received relatively little attention in comparison to insomnia [10]. Interestingly, several studies have reported that MSLT scores and objective sleep duration do not differ between those patients with depression also reporting hypersomnia versus those who do not [11–13]. There have been multiple explanations suggested for this discrepancy between subjective and objective sleepiness/sleep duration in depression. For example, that hypersomnia may be a reflection of fatigue or avolition [14], or may reflect increased time spent in bed rather than total sleep time [15]. Another possibility may be to consider hypersomnia as a subjective sleep complaint that does not require objective 'verification' of longer sleep duration. This would mirror insomnia, where no 'objective' sleep discrepancies are required for diagnosis; a minority of insomnia patients do not have any objective sleep disruption in a phenomenon known as 'paradoxical insomnia' [16]. 'Paradoxical hypersomnia' would correspondingly label individuals who feel subjectively sleepy and as if they are sleeping for long duration (even if their MSLT and sleep duration is within typical bounds).

In contrast to the mood disorder literature, hypersomnia has seldom been researched in relation to schizophrenia or related disorders. Problematic levels of excessive sleepiness were found in 32% of a group of 100 medicated patients with schizophrenia in a cross-sectional study [17]. An MSLT study of 30 patients with schizophrenia reported a higher sleep propensity in patients with psychosis compared to non-clinical controls [18]. Some studies have also reported individuals with psychosis having a higher average sleep duration than non-clinical controls [19–21], with a large proportion (43.5%-64.7%) sleeping more than 9 hours a night [22, 23].

A clear and well-known factor in excessive sleepiness in psychosis is antipsychotic medication. Feeling 'sleepy during the day' was the most commonly reported side effect of antipsychotic medication in a recent worldwide study [24]—and was associated with significant levels of distress and reduced functioning. Randomised controlled trials of medication nearly

universally identify sedation as a side-effect, although the rates vary widely (5.1%-49.1%) across different patient groups, different antipsychotic types, and different dosages [25, 26]. It is worth noting that sedation is typically assessed by self-report, rather than by using a validated measure (such as the ESS or MSLT), and 'sedation' itself is relatively poorly defined.

It is clear that antipsychotic medication contributes to excessive sleepiness in psychosis. However, there are other possible contributors to excessive sleepiness for patients with psychosis besides medication. Many other sleep problems are common in psychosis [2], and may result in excessive sleepiness due to inadequate or non-restorative sleep at night. Depression, for which hypersomnia is a core symptom, is common among those with psychosis [27]. Qualitative evidence suggests that excessive sleeping may also be used as a deliberate 'escape' strategy for some individuals [1, 28]. Notably hypersomnia has been proposed to maintain itself, via a viscous cycle wherein oversleeping leads to sleepiness due to increased sleep inertia [29], which the patient may then attempt to remedy by sleeping more [10]. These factors deserve consideration as they may indicate therapeutic approaches to target sleepiness (for example, activation experiments to test beliefs about energy and sleep, or adaptation of rise routines as described in [30]), even in spite of sedating medications.

Furthermore, excessive sleepiness could plausibly affect clinical symptoms; for example if excessive sleepiness contributes to social isolation, this could maintain paranoia by removing opportunities to gain evidence that disconfirms the threat belief [31]. As another example, if patients are feeling sleepy or tired they may be less able to ignore derogatory voices [32]. With respect to physical health, excessive sleepiness might contribute to increased sedentary behaviour [33]. Sedentary behaviour is associated with an increased risk of cardiac or metabolic health problems–and notably all-cause mortality is raised for individuals with a longer sleep duration [34]. Excessive sleeping and sleepiness may also impact on patients' ability to engage in treatment, for example by impeding attendance or engagement within mental health appointments. Given these potential negative consequences, it is surprising that there is little research on the relationship between excessive sleepiness and psychiatric symptoms or wellbeing among individuals with non-affective psychosis.

This paper reports an initial exploration of excessive sleepiness in patients with psychosis, as a first step for understanding possible causes and correlates of this clinical issue. This is based on a secondary analysis of the data collected for our paper on the incidence and severity of sleep disorders in a group of 60 outpatients with early psychosis [2]. We wanted to investigate this little researched specific type of sleep difficulty. The current investigation therefore focuses on two subgroups: those who screened positively for excessive sleep disorders (hereafter referred to as the 'excessive sleepiness' group, n = 14), and those who did not (hereafter referred to as the 'comparison' group, n = 46). The analysis focused on the following questions:

1. Medication: Does the excessive sleepiness group differ from the comparison group in medication types or dosages?

2. Sleep:

   a. Does the excessive sleepiness group have longer sleep duration, higher sleepiness, higher fatigue, or lower average activity than individuals in the comparison group?

   b. Are there sleep disorders that are more prevalent in the excessive sleepiness group versus the comparison group?

3. Psychiatric symptoms and functioning: Does the excessive sleepiness group report increased psychiatric symptoms (psychotic symptoms, depression, or anxiety) or lower quality of life than the comparison group?

## Method

### Recruitment

The current study is a secondary analysis focusing on subgroups of our previous descriptive study [2]. The study received approval from an NHS Research Ethics Committee (South West —Frenchay, reference 15/SW/0291). Participants were 18–30 years old, had a current diagnosis of non-affective psychotic disorder, were receiving care from NHS mental health teams, and were assessed as having capacity to provide informed consent by their care team and the researcher (SR). Written informed consent was provided by all participants. Patients with a primary affective disorder; primary substance abuse disorder; organic or neurological disorder; and non-fluency in English were excluded from participating in the study. For further details regarding the study population see [2].

The two study groups (excessive sleepiness and comparison) were differentiated by using the Diagnostic Interview for Sleep Patterns and Disorders (DISP; [35]). This structured interview assesses insomnia, nightmare disorder, circadian rhythm disorders, restless leg syndrome, periodic limb movement syndrome, bruxism, sleep apnea, sleep walking, night terrors, sleep enuresis, REM sleep behaviour disorder, sleep paralysis, sleep-related hallucinations, and excessive sleepiness disorders (including narcolepsy with and without cataplexy, and hypersomnia). Symptoms were rated according to diagnostic criteria, which were made as conservative as possible by including criteria from the DSM-5, ICSD-2, and ICSD-3 classification systems, and requiring that all criteria must be satisfied for a positive diagnosis to be made (see (2) for full scoring algorithm).

For the present study the excessive sleepiness group (n = 14) was defined as those screening positively for an excessive sleepiness disorder. The criteria, based on self-report, were:

1. Excessive sleepiness reported four or more days each week
   AND

2. Sleeping $\geq$ 11 hours on average in a 24 hour period (n = 5)
   OR

3. Sleep attacks (i.e. suddenly and irresistibly going to sleep) occurring four or more days each week (n = 9)

The comparison group (n = 46) were those who did not report problematic excessive sleepiness according to the above criteria in the DISP.

### Design and measures

The study is a cross-sectional between-groups comparison of the excessive sleepiness group and the comparison group.

**Demographic and clinical data.** Age, gender, and occupational status were reported by participants in the assessment. Diagnosis and medication information was collected from medical notes at the time of consent into the study. Defined daily dose (DDD; [36, 37]) and chlorpromazine equivalents [CPZ; 38] were calculated to enable comparison among differing antipsychotic medications.

**Sleep related measures.** *Subjective sleep recording*. Participants were asked to complete a Consensus Sleep Diary [39] for seven days. Average total sleep time (TST; time in bed minus time in bed spent awake) and average time spent napping during the day were calculated from the sleep diaries. Where two or fewer days of the diary were filled in the data were excluded from analysis. Of the remaining sleep diaries an average of 5.5 days were completed by participants.

*Activity*. Participants wore a wrist-based activity monitoring device (CamNTech Motion-watch 8) for seven days to allow objective recording of activity, from which sleep variables (e.g. sleep duration, efficiency) can be estimated utilising the proprietary software algorithm, although in the current study only the activity was used as an outcome variable. Actigraphic data were analysed within the Motionware software package (V1.1.25, CamNTech). Average activity was calculated across each 24 hour window for which recording was complete. The average activity count is the average number of movements per minute across that period.

*Sleepiness*. The Epworth Sleepiness Scale (ESS; 6) assessed daytime sleepiness. This scale lists eight situations. The participant rates how likely they would be to "doze off" in each situation, on a scale where 0 is "would never doze off" and 3 is "high chance of dozing off". The sum total of the score for each item is the outcome measure of the questionnaire, which can range from 0 to 24, with a score of 11 or more indicating excessive daytime sleepiness.

*Fatigue*. Fatigue was measured via the Multi-Dimensional Fatigue Symptom Inventory (MFSI) short form [40, 41]. This scale comprises 30 items in 5 subscales (General, Emotional, Physical, Mental, Vigor). Each item is rated from 0 ("Not at all") to 4 ("Extremely") by the participant. The total score for the questionnaire is the sum of the General, Emotional, Physical, and Mental subscale scores minus the Vigor subscale score. The maximum score is 100, with a higher score indicating greater fatigue.

*Comorbidity among sleep disorders*. The frequency of diagnosis or positive screen for other sleep disorders within the diagnostic interview.

**Psychiatric symptom and wellbeing measures.**   *Psychotic experiences*. Psychotic experiences were assessed with the Specific Psychotic Experiences Questionnaire [42] (SPEQ), a self-report measure assessing the past month. The subscales for paranoia, hallucinations, cognitive disorganisation, and grandiosity were used in the current study. The paranoia subscale is formed of 15 statements rated by the participant for frequency of the thought from 0 ("Not at all") up to 5 ("Nearly all the time"), with a maximum possible score of 75. The subscale for hallucinations is comprised of nine items rated on the same scale to comprise a maximum score of 45. For cognitive disorganisation eleven statements are marked as "Yes" (i.e. this is true of me) or "No" (i.e. this is not true of me), scored as 1 and 0 respectively. The grandiosity subscale is formed of eight statements rated for agreement by the participant on a 0 ("Not at all") to 3 ("Completely") scale, with a maximum score of 24. Higher scores indicate more severe psychotic experiences.

*Negative affect*. Depression and anxiety were assessed using two subscales from the Depression Anxiety and Stress Scale (DASS-21) [43, 44], a self-report measure assessing the past month. The seven items for each sub-scale are rated from 0 ("Did not apply to me at all") to 3 ("Applied to me very much"). Higher scores indicate higher levels of anxiety and depression.

*Quality of life*. Health-related quality of life was assessed with the EQ-5D-5L [45] which assesses five health related domains (mobility, self-care, ability to do usual activities, pain or discomfort, and anxiety and depression). Participants rated on a 1–5 scale their extent of issues in each domain (where 1 is "No problems", 5 is "Extreme problems"). The responses were then entered into a scoring algorithm which weighted the scoring according to domain and national population norms to give a decimal score between 0 and 1, where higher scores indicated higher health-related quality of life.

## Analysis

All analyses were carried out using SPSS 22.0 [46]. Study measures were compared between the two groups using a between-subjects t-test for dimensional measures or Chi-square for categorical tests. All testing was two-tailed, with the significance threshold set at p≤0.05. Given the small size of the groups and resulting low sensitivity (G*Power sensitivity analysis

indicated d = 1.15 as the threshold for a significant effect), confidence intervals and effect sizes were also considered as indications towards relevant relationships.

## Results

### Demographic and clinical characteristics

Demographic and clinical results can be found in Table 1. There was a higher proportion of males in the excessive sleepiness group (n = 10, 71.4%) versus the comparison group (n = 28. 60.9%). There were no differences in age and occupational status of the excessive sleepiness and comparison group. No demographic differences were statistically significant. There were also no clear differences between groups on diagnosis.

No significant differences were found in medication types or dosages between the two study groups. The small differences observed within Table 1 were in the direction of the excessive sleepiness group being marginally less likely to be prescribed antipsychotic or other medication, and at lower dosages.

### Sleep-related measures

The results from the group tests can be found in Table 2. Both study groups reported on average above 9h sleep per 24 hours, without a significant difference being found between the

**Table 1. Demographic and clinical differences between the excessive sleepiness and comparison group.**

|  | Excessive sleepiness (n = 14) | No excessive sleepiness–comparison group (n = 46) | Difference statistic[a] | p-value |
|---|---|---|---|---|
| **Gender–n (%)** |  |  | 0.515 | 0.473 |
| Male | 10 (71.4) | 28 (60.9) |  |  |
| Female | 4 (28.6) | 18 (39.1) |  |  |
| **Age–average years (SD)** | 23.57 (3.3) | 23.76 (3.3) | 0.190 | 0.850 |
| **Occupational status–n (%)** |  |  | 0.002 | 0.967 |
| Unemployed | 6 (42.9) | 20 (43.5) |  |  |
| Employed[b] | 8 (57.1) | 26 (56.6) |  |  |
| **Diagnosis—n (%)** |  |  | 2.202 | 0.699 |
| Schizophrenia | 5 (35.7) | 12 (26.1) |  |  |
| Schizoaffective disorder | 2 (14.3) | 3 (6.5) |  |  |
| Psychosis NOS | 4 (28.6) | 21 (45.7) |  |  |
| First Episode Psychosis | 3 (21.4) | 9 (19.6) |  |  |
| Schizophreniform psychosis | 0 (0.0) | 1 (2.2) |  |  |
| **Medications prescribed–n (%)** |  |  | 0.554 | 0.758 |
| Antipsychotic | 10 (71.4) | 39 (84.8) |  |  |
| Anti-depressant | 6 (42.9) | 16 (34.8) |  |  |
| Anxiolytic | 0 | 1 (2.2) |  |  |
| Mood stabilizer | 0 | 2 (4.3) |  |  |
| Hypnotic | 0 | 3 (6.5) |  |  |
| No medication | 2 (14.3) | 5 (10.9) |  |  |
| **Antipsychotic dosage–mean (SD)[c]** |  |  |  |  |
| Defined Daily Dose (DDD) | 0.96 (0.7) | 1.07 (0.7) | 0.456 | 0.651 |
| Chlorpromazine equivalent (CPZ) | 285.0 (397.5) | 317.5 (208.1) | 0.354 | 0.725 |

[a]Chi-square statistic for gender, occupational status, diagnosis and medications prescribed, t-statistic for age and antipsychotic dose.

[b]Includes part time; part time (sick leave); full time; full time (sick leave); student; volunteer; home-maker.

[c]Statistics only include participants prescribed antipsychotic medication i.e. n = 10 in excessive sleepiness group, n = 39 in comparison group.

**Table 2. Sleep-related differences between the excessive sleepiness and comparison groups.**

| | Excessive sleepiness | No excessive sleepiness–comparison group | t-statistic | p-value | Effect size (d) | 95% Confidence Interval | |
|---|---|---|---|---|---|---|---|
| | Mean (SD) | Mean (SD) | | | | Lower | Upper |
| Average sleep duration in 24hrs | 9h42m (1h46m) | 9h29m (1h50m) | 3.92 | 0.696 | 0.12 | -54m | 1h20m |
| Average activity in 24hrs | 104.30 (52.5) | 150.99 (82.9) | -2.510 | 0.017 | -0.59 | -84.47 | -8.91 |
| Sleepiness (ESS)[a] | 10.63 (4.9) | 7.22 (3.7) | 2.054 | 0.049 | 0.80 | 0.01 | 6.80 |
| Fatigue (MFSI) | 38.00 (25.8) | 30.32 (23.6) | 1.039 | 0.303 | 0.32 | -7.13 | 22.39 |

[a]n = 28 cases.

ESS = Epworth Sleepiness Scale.

MFSI = Multi-dimensional Fatigue Symptom Inventory.

groups. Activity recorded in the excessive sleepiness group was significantly lower than in the comparison group. The excessive sleepiness group scored significantly higher on the ESS scale, although again both groups reported a high average level of sleepiness (>7 on the ESS). Fatigue levels appeared higher on average in the excessive sleepiness group, however the difference was not significant.

Excessive sleepiness had a high level of comorbidity with other sleep disorders (see Table 3). Only two individuals screened positively for excessive sleepiness with no other sleep problems reported (16.6%), with the remaining 12 screening positively for insomnia (n = 10, 83.3%), nightmare disorder (n = 9, 75.0%), and a variety of other sleep disorders. Comorbidity was extremely high among all sleep disorders, with an average of 2.8 additional sleep disorders per individual in the excessive sleepiness group. No significant differences were found in rates of individual sleep disorders between the excessive sleepiness group and the comparison group.

## Psychiatric symptom measures

Table 4 contains the results from the between-groups comparison of symptom measures. No significant differences were found between the study groups on any psychiatric symptom or general functioning measures. Inspection of confidence intervals and mean differences

**Table 3. Comorbid sleep disorders in excessive sleepiness and comparison groups.**

| | Excessive sleepiness | No excessive sleepiness–comparison group |
|---|---|---|
| Disorder | Frequency–n (%) | Frequency–n (%) |
| **No comorbid sleep disorders** | 2 (14.3) | 12 (26.1) |
| **Comorbid sleep disorders** | | |
| Insomnia | 10 (71.4) | 20 (43.4) |
| Nightmare disorder | 9 (64.3) | 20 (43.4) |
| Sleep-related hallucinations | 5 (35.7) | 20 (43.4) |
| PLM | 4 (28.6) | 8 (17.3) |
| RLS | 3 (21.4) | 11 (23.9) |
| Sleep paralysis | 3 (21.4) | 6 (13.0) |
| Circadian disorder | 3 (21.4) | 2 (4.3) |
| Night terror | 2 (14.3) | 3 (6.5) |

NB Percentages add up to more than 100% due to multiple comorbidities among sleep disorders.

No differences reached significance (p≤0.05) in chi-square test.

**Table 4. Symptom differences between the excessive sleepiness and comparison group.**

| Symptom | Excessive sleepiness | No excessive sleepiness–comparison group Mean (SD) | t-statistic | p-value | Effect size (d) | 95% Confidence Interval | |
|---|---|---|---|---|---|---|---|
| | Mean (SD) | | | | | Lower | Upper |
| **Psychotic experiences (SPEQ)** | | | | | | | |
| Paranoia | 21.42 (18.6) | 32.46 (22.8) | -1.647 | 0.105 | -0.50 | -24.43 | 2.37 |
| Hallucinations | 11.64 (10.3) | 15.43 (12.2) | -1.051 | 0.298 | -0.32 | -11.01 | 3.42 |
| Cognitive Disorganisation | 7.86 (2.6) | 6.02 (3.3) | 1.927 | 0.059 | 0.58 | -0.71 | 3.74 |
| Grandiosity | 10.00 (7.2) | 7.02 (5.7) | 1.604 | 0.114 | 0.48 | -0.74 | 6.69 |
| **Negative affect (DASS)** | | | | | | | |
| Depression | 9.43 (6.0) | 8.30 (5.1) | 0.696 | 0.489 | 0.21 | -2.11 | 4.36 |
| Anxiety | 10.29 (5.3) | 9.30 (6.6) | 0.510 | 0.612 | 0.16 | -2.87 | 4.84 |
| **Wellbeing (EQ-5D)** | | | | | | | |
| Health related quality of life | 0.59 (0.2) | 0.64 (0.2) | -0.857 | 0.395 | -0.25 | -0.20 | 0.08 |

indicates that self-reported paranoia and hallucinations are less severe among those reporting excessive sleepiness versus the comparison group. However, self-reported cognitive disorganization and grandiosity are raised in the excessive sleepiness group compared to patients without excessive sleepiness. Negative affect (depression and anxiety) and health related quality of life were similar in both groups, but in these cases the differences seen tended towards greater severity in the excessive sleepiness group rather than the comparison group.

## Discussion

Excessive sleepiness is under-researched in patients with psychosis but has the potential to be of clinical consequence due to the disruption caused, and its physical and mental health effects. This exploratory analysis aimed to provide initial data on the character and correlates of excessive sleepiness in psychosis by investigating differences between psychosis patients with excessive sleepiness and with those without excessive sleepiness. Surprisingly no significant differences were found in medication or dosage between the two study groups. The excessive sleepiness group reported significantly lower average activity than comparison group, and significantly higher levels of sleepiness on the Epworth sleepiness scale. However, subjectively reported sleep duration did not differ between the study groups, although both groups did report extended sleep duration (>9h). There was high comorbidity of sleep disorders with excessive sleepiness–in particular insomnia, nightmares, and sleep-related movement disorders. Regarding clinical symptoms, cognitive disorganization was more severe in the excessive sleepiness group, but the data also indicated lower severity of paranoia and hallucinations (although none of these differences were significant). The findings also indicated higher levels of depression and anxiety, and reduced quality of life, in those reporting excessive sleepiness, but again none of these differences were statistically significant. Overall, the results indicate that excessive sleepiness is not solely explicable by medication factors, here instead being linked to overall activity levels and presence of sleep disorders. This is a novel finding that, if replicated, may have implications for clinical intervention. The results also indicate a mixed profile in psychiatric symptoms among patients with excessive sleepiness, with some dimensions appearing more severe (cognitive disorganization, grandiosity) and others less (paranoia, hallucinations). These associations require further investigation to characterize these associations and clarify the direction of their relationships with excessive sleepiness and other clinical factors (e.g. negative affect).

The lack of association between antipsychotic medication dose and excessive sleepiness was unexpected. However, one possible explanation could be that all participants were in fact sedated from medications. In support of this, both groups had longer than average sleep durations (average would be approximately eight hours for this age group; [47]). Furthermore. one hundred or fewer counts of activity per minute has been categorised as sedentary behaviour [48], and both groups were within one standard deviation of this cut off. Lastly both groups showed scores on upper bounds (>7) of previously published norms of the Epworth sleepiness scale (4.5±3.3 in non clinical population [49]). Regardless, it appears that sedation from antipsychotic medication alone is not the sole explanatory factor in problematic excessive sleepiness among those with psychosis.

The relationships between excessive sleepiness and clinical symptoms present an interesting picture. It is important not to overinterpret the findings given the lack of significant differences. Nevertheless, the picture emerging suggests less severe paranoia and hallucinations and more severe cognitive disorganisation and grandiosity as being associated with excessive sleepiness, which is an interesting clinical profile to consider. It could be the case that sleepiness and reduced paranoia are both related to reduced hypervigilance, but this is inconsistent with the similar anxiety levels in both groups. It is also possible that the excessive sleepiness group are less active and therefore encountering less stressful situations, or using sleep successfully as an 'escape' as suggested previously [28]. Overall, the relationship between sleepiness and clinical presentation in psychosis deserves further investigation, which should include investigation of possible 'paradoxical hypersomnia' i.e. patient reports of sleepiness and long sleep times as reflecting fatigue or depression rather than objective sleep differences. This is especially important given the low rate of clinical attention; 11 out of 14 (78.6%) of the excessive sleepiness group reported no clinical treatment of their sleepiness [2].

## Limitations

This was an exploratory investigation of a common clinical observation. Although novel, it has many limitations. One limitation is a lack of direct hypotheses that could be made regarding the impact of excessive sleepiness. In order to direct the analyses in this paper, particular areas of interest have been identified from the relevant literatures on hypersomnia, mood disorder, and sedation in psychosis. Another limitation is that the results were not corrected for multiple testing; therefore they should be interpreted with care. The study was cross-sectional, therefore it is not possible to investigate direction of influence between excessive sleepiness and any other factors. As with our previous study [2], the lack of polysomnography limits our ability to investigate objective sleep differences between our study groups; this would be especially relevant given the possibility of 'paradoxical hypersomnia'. Another consideration is the use of defined daily dose (DDD) and chlorpromazine (CPZ) equivalents for our analyses. These comparison systems are based on antipsychotic effects (i.e. dopaminergic blockade), and not on sedative properties–which are thought to occur via histaminergic interaction, among other routes–and therefore have limitations in applicability for our investigation.

It is worth noting some issues in measuring excessive sleepiness in individuals with psychosis, based on observations during the study. First, the DISP asks if sleepiness has intruded with school, work, or other activities. Negative responses to this question in the 'no sleep disorder' group may therefore be confounded by low daytime activity itself (i.e. there are limited activities for the sleepiness to interfere with). Second, naps may have been under-reported in the sleep diary due to a lack of nap detail requested by the consensus sleep diary [39], particularly when recording multiple naps during the day or unsuccessful nap attempts. Lastly, individual ESS scale items, particularly those involving dozing off around other people or in public places,

may have not been endorsed due to participants' hypervigilance in these situations. The small sample size precluded item response analyses to investigate this issue further, but this could have induced a ceiling effect in ESS scores. These issues support the need to develop specific conceptualisation and measurement of excessive sleepiness in psychosis, which would be an important area for service user involvement to identify how they experience oversleeping or daytime sleepiness, and what the consequences of this are for them.

In conclusion, this is the first study to investigate the character and correlates of excessive sleepiness in psychosis. This in itself is surprising given the frequency with which sedation, sleepiness, and oversleeping are identified as issues by patients and clinicians [3, 24, 25]. One particular focus might be to develop interventions (for example, behavioural activity interventions as described in [30]) to test their effect on excessive sleepiness in this patient group, as these results suggest improvement may be possible despite the presence of sedating medications. These issues deserve further investigation, ideally with longitudinal or experimental study designs to establish the direction of the effects amongst mood, sleep disorders, activity, medication, and psychotic symptoms.

## Author Contributions

**Conceptualization:** Sarah Reeve, Bryony Sheaves, Daniel Freeman.

**Formal analysis:** Sarah Reeve.

**Investigation:** Sarah Reeve.

**Methodology:** Sarah Reeve, Bryony Sheaves, Daniel Freeman.

**Supervision:** Bryony Sheaves, Daniel Freeman.

**Writing – original draft:** Sarah Reeve.

**Writing – review & editing:** Sarah Reeve, Bryony Sheaves, Daniel Freeman.

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
