## [Decision Letter · Decision Letter 0]

17 Nov 2020

PONE-D-20-28283

Excessive sleepiness in patients with psychosis: an initial investigation

PLOS ONE

Dear Dr. Reeve,

Thank you for submitting your manuscript to PLOS ONE. After careful consideration, we feel that it has merit but does not fully meet PLOS ONE’s publication criteria as it currently stands. Therefore, we invite you to submit a revised version of the manuscript that addresses the points raised during the review process (please see below).

We look forward to receiving your revised manuscript.

Kind regards,

Federica Provini

Academic Editor

PLOS ONE

2. Please describe in your methods section how capacity to consent was determined for the participants in this study.

3.We note that you have indicated that data from this study are available upon request. PLOS only allows data to be available upon request if there are legal or ethical restrictions on sharing data publicly. For information on unacceptable data access restrictions, please see http://journals.plos.org/plosone/s/data-availability#loc-unacceptable-data-access-restrictions.

Reviewers' comments:

Reviewer #1: In this work, Reeve and colleagues successfully tried to pave the basis for studying the mechanisms of a common phenomenon seen in the clinical practice, namely, the significantly high amount of sleepiness in patient with psychosis. They compared two previously reported groups of psychotic patients with similar diagnosis: one group suffering from excessive sleepiness, the other not. They focused on differences among therapies, quality of sleep, fatigue, average activity, prevalence of sleep disorders, psychiatric symptoms, and quality of life.

The work does not show significant differences in any comparisons, but in the sleep-related ones. Specifically, significant differences are shown for 24h average activity, sleepiness amount and fatigue (table 2), confirming the correct separation of the two groups.

The conclusions reported by the authors are in line with their findings and reproducible. Moreover, the sample size and statistical power reach the threshold accepted by the community.

The language of the manuscript is clear, effective, and intelligible. Nevertheless, showing visual graphic for the results may help its fruition.

Minor revisions:

The authors propose to mirror the paradoxical insomnia to sleepiness for avoiding objective verification of hypersomnia. Therefore, they should call their sleepiness paradoxical hypersomnia. Moreover, despite hypersomnia, as described in introduction, can be considered different (correctly) than sleepiness, the two terms are used as equivalent in the discussion.

Introduction:

Reeve and colleagues do not refer to any work for the definition of sleepiness. Is this their own definition? If so, it should be stated. The paragraph regarding contributors to excessive sleepiness can have a clearer logical flow. One space is needed before the sentence next to ref 31 (moreover, ref 31 should have been numbered within the previous paragraph). The comparison group is referred as ‘no excessive sleepiness’ group (probably worth to be used here as well) in the results rather than ‘comparison’ group as stated in this section.

Methods:

Recruitment: References or reasons for the criteria chosen should be added (or re-referenced).

Subjective sleep recording: The authors should report the distinct number of days filled by each participant.

Psychotic experience: The term ‘past month’ can be misleading. Quotation marks should be moved after brackets within the sentence about cognitive disorganization.

Negative affect: The term ‘past month’ can be misleading. Quotation marks should be consistent with the previous paragraph.

Quality of life: Verb tense inconsistency is present within this paragraph.

Analysis: Verb tense inconsistency is present within this paragraph. Furthermore, a comma should be added after the last bracket.

Demographic: 70,4% can be considered more than slightly higher proportion of males. Here ‘comparison’ (from introduction) and ‘no excessive sleepiness’ (quotation mark should be added) terms are considered equivalent without previous clarifications.

Table 1: Occupational status information does not seem relevant for the rest of this manuscript. However, if cited, it may be interesting for the second paragraph of the ‘Limitations’ section.

Table 2: SD of comparison group average sleep duration should fit on the upper line. Fatigue’ p-value results difficult to interpret. The caption should explain the acronyms used (not only in this table, otherwise please remove them).

Discussion:

Reeve and colleagues have shown no statistically significant differences between ‘symptoms’ among the two groups. However, some interpretations about comorbidities strongly highlight differences and this might be seen as ambiguous. For example, would be worth to remove from brackets ‘’although these differences were not significant’’ from the following sentence:

‘’Regarding clinical symptoms, cognitive disorganization was more severe in the excessive sleepiness group, but the data also indicated lower severity of paranoia and hallucinations (although these differences were not significant).’’

Maslowsky & Ozer 2014 and Parkes et al., 1998 references should be formatted according to the rest of the document.

Reviewer #2: The proposed manuscript investigates an important and treatable clinical issue in non-affective psychosis. Hypersomnolence is seldom investigated and formally assessed in clinical practice, and remains an under-researched topic in this category of patients. Furthermore, hypersomnolence is not homogeneously defined in available studies, leading to highly heterogeneous findings.

The authors present a secondary analysis of a previous descriptive study (“Sleep Disorders in Early Psychosis: Incidence, Severity, and Association With Clinical Symptoms”), focusing on subgroups differentiated by using the Diagnostic Interview for Sleep Patterns and Disorders. In a cross-sectional study, psychotic outpatients reporting excessive sleepiness were compared with psychotic patients who did not report excessive sleepiness,. The authors found significantly lower average activity and significantly higher levels of sleepiness on the ESS in the excessive sleepiness group. No significant differences were found in medication or dosage, psychiatric symptoms and sleep duration between the two study groups.

The study design has several limitations, which have been clearly discussed in the dedicated section: small sample size, absence of correction for multiple testing, wide confidence intervals, etc. Overall, the paper is well-written, methods are detailed and clear, the authors provided hypotheses to explain the results and they suggested future perspectives of improvements.

I think that a more elaborated discussion is needed regarding the following issues:

-The authors mentioned organic and neurological disease among the exclusion criteria. Did any of the patients take non-psychiatric medications? Could they interfere with sleep or cause sleepiness?

-Was the severity of psychotic symptoms assessed with objective measurements in addition to self-reported questionnaires?

-The authors found no differences between the groups in antipsychotic medication type and dosage using Defined Daily Dose and Chlorpromazine Equivalents for comparisons. These systems have some intrinsic limitations when investigating sedation instead of antipsychotic effects. Antipsychotic medications have variable sedative effects. Chlorpromazine Equivalents are based primarily on dopaminergic blockade and not upon a drug receptor profile for histaminergic systems, among others. I think this could be mentioned in the “limitations” section.

-Some of the limitations described in the previous study could be mentioned in the new paper in the dedicated section, for example the lack of polysomnography (which could provide useful data to make a differential diagnosis of multiple causes of excessive sleepiness) and the issue of representativeness of the participant group.

-Was excessive sleepiness discussed, formally assessed and treated in clinical practice in these patients? This aspect was explored in the previous study; I think it would be useful to also mention this in the new manuscript.

-The authors found hallucinations to be less severe in the excessive sleepiness group, although results were not statistically significant. Were sleep-related hallucinations investigated or discriminated from day-time hallucinations?

-The core take-home message of the manuscript is that more attention should be given to factors influencing excessive sleepiness other than medication in psychotic patients. The manuscript is based on data from a previous study with different objectives, but despite its limitations it offers stimulating new data that suggest new perspectives for further investigations.

---

## [Author Response · Author response to Decision Letter 0]

4 Dec 2020

We would like to thank the reviewers for their positive comments and constructive suggestions for our manuscript. We believe it has been greatly improved by the changes suggested. Please see below for our responses to each comment. 

Reviewer #1: 

1) In this work, Reeve and colleagues successfully tried to pave the basis for studying the mechanisms of a common phenomenon seen in the clinical practice, namely, the significantly high amount of sleepiness in patient with psychosis. They compared two previously reported groups of psychotic patients with similar diagnosis: one group suffering from excessive sleepiness, the other not. They focused on differences among therapies, quality of sleep, fatigue, average activity, prevalence of sleep disorders, psychiatric symptoms, and quality of life.

The work does not show significant differences in any comparisons, but in the sleep-related ones. Specifically, significant differences are shown for 24h average activity, sleepiness amount and fatigue (table 2), confirming the correct separation of the two groups.

The conclusions reported by the authors are in line with their findings and reproducible. Moreover, the sample size and statistical power reach the threshold accepted by the community.

The language of the manuscript is clear, effective, and intelligible. 

Thank you for these positive comments on our manuscript. 

Minor revisions:

2) The authors propose to mirror the paradoxical insomnia to sleepiness for avoiding objective verification of hypersomnia. Therefore, they should call their sleepiness paradoxical hypersomnia. Moreover, despite hypersomnia, as described in introduction, can be considered different (correctly) than sleepiness, the two terms are used as equivalent in the discussion.

Thank you for noting this – we have added a mention of ‘paradoxical hypersomnia’ in the introduction and discussion as part of the interpretation of the results, and amended other mentions of hypersomnia in the discussion to correctly reflect ‘sleepiness’ as you state. 

Introduction:

3) Reeve and colleagues do not refer to any work for the definition of sleepiness. Is this their own definition? If so, it should be stated. 

We have added an appropriate reference for this definition on p3, thank you for pointing this out. 

4) The paragraph regarding contributors to excessive sleepiness can have a clearer logical flow. One space is needed before the sentence next to ref 31 (moreover, ref 31 should have been numbered within the previous paragraph). 

We have adjusted the order of sentences (and adjusted the reference) in this paragraph (p4) and believe it reads more clearly now, thank you for noting this. 

5) The comparison group is referred as ‘no excessive sleepiness’ group (probably worth to be used here as well) in the results rather than ‘comparison’ group as stated in this section.

Thank you for noticing this. We have elected to confirm the wording throughout as comparison group. While we realise ‘no excessive sleepiness’ is in some ways clearer on its own, we find that within sentences it becomes unclear and laborious for the reader hence our usage of ‘comparison’ while clarifying (both prior to results and throughout in the tables) that the comparison group is the ‘no excessive sleepiness’ group. 

Methods:

6) Recruitment: References or reasons for the criteria chosen should be added (or re-referenced).

We have elected to re-reference readers towards our previous descriptive study which provides further rationale for e.g. age of participants.

7) Subjective sleep recording: The authors should report the distinct number of days filled by each participant.

The average number of days completed in was 5.5 across the study group (after excluding those filled in for 2 or less), this figure has been added into the methods.

8) Psychotic experience: The term ‘past month’ can be misleading. Quotation marks should be moved after brackets within the sentence about cognitive disorganization.

The questionnaire does ask for experiences ‘over the past month’ and therefore we think this is a fair statement to make in describing the questionnaire. The description of the cognitive disorganisation responses has been amended to clarify further- it is not appropriate to include the bracketed phrases in the quotation marks as the questionnaire response text only includes “Yes” and “No”. 

9) Negative affect: The term ‘past month’ can be misleading. Quotation marks should be consistent with the previous paragraph.

Similarly to previous, we appreciate this point but given the questionnaire itself asks for ‘past month’ we think this is a fair representation of the measure. 

10) Quality of life: Verb tense inconsistency is present within this paragraph.

We have amended the grammar accordingly. 

11) Analysis: Verb tense inconsistency is present within this paragraph. Furthermore, a comma should be added after the last bracket.

These changes have been made, thank you for noticing these errors. 

12) Demographic: 70,4% can be considered more than slightly higher proportion of males. Here ‘comparison’ (from introduction) and ‘no excessive sleepiness’ (quotation mark should be added) terms are considered equivalent without previous clarifications.

We have removed ‘slightly’, and amended the comparison group reference as according to previous comment. 

13) Table 1: Occupational status information does not seem relevant for the rest of this manuscript. However, if cited, it may be interesting for the second paragraph of the ‘Limitations’ section.

Occupational status is provided as part of our standard demographic questionnaire. Unfortunately the interpretation of this as regards the limitations noted is restricted as ‘employed’ includes those who were on sick leave - therefore functionally are unlikely to differ from those who were unemployed in terms of daily activity in the study period. 

14) Table 2: SD of comparison group average sleep duration should fit on the upper line. Fatigue’ p-value results difficult to interpret. The caption should explain the acronyms used (not only in this table, otherwise please remove them).

We have adjusted the relevant column widths (although any issue with this that remain would be resolved in proofing if accepted) and added clarification on the acronyms used. We are unclear on if the comment re: fatigue’s p-value relates to its presentation in the table or the content itself – re the latter, we agree that it is interesting in the context of ‘paradoxical hypersomnia’ proposed above. However, as with sleep duration both study groups are reporting relatively high levels of fatigue and therefore it fits within our general interpretation of the results. 

Discussion:

15) Reeve and colleagues have shown no statistically significant differences between ‘symptoms’ among the two groups. However, some interpretations about comorbidities strongly highlight differences and this might be seen as ambiguous. For example, would be worth to remove from brackets ‘’although these differences were not significant’’ from the following sentence:

‘’Regarding clinical symptoms, cognitive disorganization was more severe in the excessive sleepiness group, but the data also indicated lower severity of paranoia and hallucinations (although these differences were not significant).’’

We have clarified this sentence to reduce the ambiguity thank you for noticing this potential confusion.

16) Maslowsky & Ozer 2014 and Parkes et al., 1998 references should be formatted according to the rest of the document.

We have amended these references, thank you for noticing this issue. 

Reviewer #2: 

1) The proposed manuscript investigates an important and treatable clinical issue in non-affective psychosis. Hypersomnolence is seldom investigated and formally assessed in clinical practice, and remains an under-researched topic in this category of patients. Furthermore, hypersomnolence is not homogeneously defined in available studies, leading to highly heterogeneous findings.

The authors present a secondary analysis of a previous descriptive study (“Sleep Disorders in Early Psychosis: Incidence, Severity, and Association With Clinical Symptoms”), focusing on subgroups differentiated by using the Diagnostic Interview for Sleep Patterns and Disorders. In a cross-sectional study, psychotic outpatients reporting excessive sleepiness were compared with psychotic patients who did not report excessive sleepiness,. The authors found significantly lower average activity and significantly higher levels of sleepiness on the ESS in the excessive sleepiness group. No significant differences were found in medication or dosage, psychiatric symptoms and sleep duration between the two study groups.

The study design has several limitations, which have been clearly discussed in the dedicated section: small sample size, absence of correction for multiple testing, wide confidence intervals, etc. Overall, the paper is well-written, methods are detailed and clear, the authors provided hypotheses to explain the results and they suggested future perspectives of improvements.

Thank you for these positive comments, we agree it is an important but under-researched area. 

2) I think that a more elaborated discussion is needed regarding the following issues:

The authors mentioned organic and neurological disease among the exclusion criteria. Did any of the patients take non-psychiatric medications? Could they interfere with sleep or cause sleepiness?

Unfortunately, we do not have data available on this issue – we only have medication data relating to the categories listed. Our impression is that there would not be enough individuals being prescribed any other particular set of medications to form a meaningful comparison (given the massive range of other medications/combinations that might be prescribed). Furthermore, while many medications have sedation as a side effect, the magnitude of the expected effect is not clear, let alone when they are combined with other medications. Therefore, we think it is appropriate for our study to have restrained itself to the medications commonly prescribed in the study population – most importantly of course, antipsychotic medications.

3) Was the severity of psychotic symptoms assessed with objective measurements in addition to self-reported questionnaires?

No, only the self-reported questionnaires reported were used in the study. 

4) The authors found no differences between the groups in antipsychotic medication type and dosage using Defined Daily Dose and Chlorpromazine Equivalents for comparisons. These systems have some intrinsic limitations when investigating sedation instead of antipsychotic effects. Antipsychotic medications have variable sedative effects. Chlorpromazine Equivalents are based primarily on dopaminergic blockade and not upon a drug receptor profile for histaminergic systems, among others. I think this could be mentioned in the “limitations” section.

This is a very good point and we have included it in the limitations (p15); thank you for this comment. 

5) Some of the limitations described in the previous study could be mentioned in the new paper in the dedicated section, for example the lack of polysomnography (which could provide useful data to make a differential diagnosis of multiple causes of excessive sleepiness) and the issue of representativeness of the participant group.

Thank you for noting this. We have added a sentence to reflect the lack of PSG in the limitations, as the comments re paradoxical hypersomnia from the previous reviewer have made this especially salient. Given the difference in aims to our previous study (which was reporting on prevalence of sleep disorders, whereas the current study is comparison based), we do not think the representativeness is as key an issue, but we are willing to add it in as well if further advised. 

6) Was excessive sleepiness discussed, formally assessed and treated in clinical practice in these patients? This aspect was explored in the previous study; I think it would be useful to also mention this in the new manuscript.

This is another very good point; we have added a sentence with this context into the discussion (p14; 11 out of 14 of the excessive sleepiness group reported no treatment of their sleepiness). 

7) The authors found hallucinations to be less severe in the excessive sleepiness group, although results were not statistically significant. Were sleep-related hallucinations investigated or discriminated from day-time hallucinations?

This is an interesting question – the hallucinations scale from the SPEQ does not specifically exclude sleep-related hallucinations, and therefore could be confounded by these experiences. As part of the wider study we assessed presence of sleep-related hallucinations, which were reported by 5 of the excessive sleepiness group (35.7%) versus 20 of the no excessive sleepiness group (43.5%). Therefore the higher level of hallucinations in the comparison group could theoretically in part be attributable to an increased rate of sleep-related hallucinations. However, the reverse confound could also be the case (i.e. the sleep-related hallucinations could be psychotic hallucinations occurring at sleep-onset) – there is little guidance on discriminating these phenomena as highlighted in our previous paper. Given that resulting un-interpretability of any overlap, the complexity of the issue, and its tangential relationship to sleepiness (which is the focus of the current paper), we have not amended the manuscript to include this debate, but if the editor or reviewer feels it is worth mention we would be willing to revisit to add these issues to the discussion.

8) The core take-home message of the manuscript is that more attention should be given to factors influencing excessive sleepiness other than medication in psychotic patients. The manuscript is based on data from a previous study with different objectives, but despite its limitations it offers stimulating new data that suggest new perspectives for further investigations.

Thank you for this comment, we agree and are happy to receive this feedback.

---

## [Decision Letter · Decision Letter 1]

26 Dec 2020

Excessive sleepiness in patients with psychosis: an initial investigation

PONE-D-20-28283R1

Dear Dr. Reeve,

We’re pleased to inform you that your manuscript has been judged scientifically suitable for publication and will be formally accepted for publication once it meets all outstanding technical requirements.

Kind regards,

Federica Provini

Academic Editor

PLOS ONE

Reviewer's Responses to Questions

**Comments to the Author**

1. If the authors have adequately addressed your comments raised in a previous round of review and you feel that this manuscript is now acceptable for publication, you may indicate that here to bypass the “Comments to the Author” section, enter your conflict of interest statement in the “Confidential to Editor” section, and submit your "Accept" recommendation.

Reviewer #1: All comments have been addressed

Reviewer #2: All comments have been addressed

2. Is the manuscript technically sound, and do the data support the conclusions?

Reviewer #1: Yes

Reviewer #2: Yes

3. Has the statistical analysis been performed appropriately and rigorously? 

Reviewer #1: Yes

Reviewer #2: Yes

4. Have the authors made all data underlying the findings in their manuscript fully available?

Reviewer #1: No

Reviewer #2: Yes

5. Is the manuscript presented in an intelligible fashion and written in standard English?

Reviewer #1: Yes

Reviewer #2: Yes

6. Review Comments to the Author

Reviewer #1: (No Response)

Reviewer #2: (No Response)

7. PLOS authors have the option to publish the peer review history of their article (what does this mean?). If published, this will include your full peer review and any attached files.

Reviewer #1: No

Reviewer #2: **Yes: **Armando D'Agostino

---

## [Editor Report · Acceptance letter]

4 Jan 2021

PONE-D-20-28283R1 

Excessive sleepiness in patients with psychosis: an initial investigation 

Dear Dr. Reeve:

I'm pleased to inform you that your manuscript has been deemed suitable for publication in PLOS ONE. Congratulations! Your manuscript is now with our production department. 

Kind regards, 

on behalf of

Dr. Federica Provini 

Academic Editor

PLOS ONE